# Counterfactual thinking in psychiatric and neurological diseases: A scoping review

Sofia Tagini[1]*, Federica Solca[2], Silvia Torre[1], Agostino Brugnera[3], Andrea Ciammola[1], Ketti Mazzocco[4,5], Roberta Ferrucci[6,7,8], Vincenzo Silani[1,2], Gabriella Pravettoni[4,5], Barbara Poletti[1]

1 Department of Neurology and Laboratory of Neuroscience—IRCCS Istituto Auxologico Italiano, Milan, Italy, 2 Department of Pathophysiology and Transplantation, "Dino Ferrari" Center, University of Milan, Milan, Italy, 3 Department of Human and Social Sciences, University of Bergamo, Bergamo, Italy, 4 Department of Oncology and Hemato-Oncology, University of Milan, Milan, Italy, 5 Applied Research Division on Cognitive and Psychological Sciences, European Institute of Oncology, Milan, Italy, 6 Department of Health Sciences, Aldo Ravelli Center for Neurotechnology and Experimental Brain Therapeutics, International Medical School, University of Milan, Milan, Italy, 7 ASST Santi Paolo e Carlo, Neurology Clinic III, Milan, Italy, 8 IRCCS Ca' Granda Foundation Maggiore Policlinico Hospital, Milan, Italy

* tagini.sofia@gmail.com

## Abstract

### Background

The ability to simulate alternatives to factual events is called counterfactual thinking (CFT) and it is involved both in emotional and behavioral regulation. CFT deficits have been reported in psychiatric and neurological conditions, possibly contributing to patients' difficulties in modulating behaviors and affections. Thus, acknowledging the presence and possible consequences of CFT impairments might be essential for optimal clinical management.

### Objectives

This scoping review aims to summarize the previous evidence about CFT in psychiatric and neurological diseases to determine the extent of the previous research and what has been discovered so far, the variety of clinical conditions considered, the methodologies adopted, and the relevant issues to be addressed by future investigations.

### Methods

PsycInfo, PubMed, Scopus, and Web of Science were searched to identify articles published up to January 2020, written in English and focused on CFT in adults affected by psychiatric or neurological conditions.

### Results

Twenty-nine studies have been included; most of them focused on psychiatric conditions, a minority considered neurological diseases. The generation of counterfactual thoughts related to a negative real-life or a fictional event and the counterfactual inference test were the most popular tasks adopted. CFT impairments were reported in both psychiatric and neurological conditions, likely associated with a fronto-executive dysfunction.

**Data Availability Statement:** All relevant data are within the manuscript and its Supporting Information files.

**Funding:** The authors received no specific funding for this work.

**Competing interests:** The authors have declared that no competing interests exist.

## Conclusions

Future research might further explore CFT in those psychiatric and neurological conditions in which CFT difficulties have been preliminary reported. Furthermore, it would be recommendable to extend this investigation to all the clinical conditions possibly at risk of fronto-executive dysfunction. In the end, we speculate that since CFT plays a role in driving everyday behaviors, it might be crucial also when medical decisions are involved; thus, future research might extend the investigation of CFT especially to those populations that implicate complex clinical management.

## Introduction

The properness of our past actions is judged according to counterfactual possibilities; that is, what could have been if we had behaved differently. The ability to simulate alternatives to factual events is called counterfactual thinking (CFT), and it is a signature of humans' cognition [1]. Conditional propositions are typically used to think counterfactually, often converting unusual behaviors into more "normal" antecedents [1]: "If I had taken the usual way, I would not have missed the flight". The closer the outcome is to the expected (missed) goal, the higher the probability to elicit CFT [1].

But why do we think counterfactually? The acknowledgement that a better (upward) outcome could have been achieved elicits regret, guilty, and self-blame; the greater the salience of the missed achievement, the worse the judgment [2, 3]. Nevertheless, we might learn an important lesson for the next time. Indeed, the most prominent function of upward CFT is to provide useful insight about possible alternative ways to achieve missed goals, influencing future behaviors [4, 5]; obviously, as long as we recognize that what we did (or did not) was the causal antecedent of our failure. On the contrary, downward counterfactuals denote worse possible scenarios, prompting a sense of relief, thus, having a beneficial affective-regulation purpose [6].

Consequently, impaired CFT might have deleterious effects on the ability to modulate both behaviors and affections. Lack of CFT could impact behavioral regulation preventing individuals' experiential learning and determining underachievement and/or social dysfunction; conversely, excessive CFT may lead to unnecessary worries, anxiety, and dysphoria [4, 5]. Thus, CFT difficulties might be somehow related to all those clinical conditions characterized by intense psychological distress, and emotional or behavioral dysfunctions. In support of this hypothesis, meta-analyses showed that there is a significant association between CFT hyperactivation and post-traumatic stress disorder (PTSD) [7] and between upwards counterfactuals generation and depression [8]. Furthermore, altered CFT was observed in other psychiatric conditions, such as schizophrenia [9, 10] and the obsessive compulsive disorder (OCD) [11]. The causal direction of these associations and the role of possible moderating factors are still under debate [4, 7, 8]. On the other hand, the literature suggests quite strongly that CFT difficulties might be related to fronto-executive vulnerabilities, at least concerning depression and schizophrenia [4, 12, 13]. Indeed, CFT impairments have been observed also in neurological conditions characterized by a fronto executive dysfunction, such as Parkinson's Disease (PD) [14], Huntington's Disease (HD) [15], and frontal lobe damages [16].

To sum up, CFT difficulties may be detected in a wide range of psychiatric and neurological conditions, with possible negative effects on people's affective and behavioral regulation. Accordingly, the acknowledgement of the clinical populations possibly involved might be

essential for optimal assistance and proper clinical management. For instance, a more refined understanding of the possible role of counterfactuals in maintaining PTSD and depressive symptoms might be crucial for tailoring efficacious interventions [7], which could address self-compassion and aim at minimizing the contemplation of negative outcomes [8]. Furthermore, in those conditions characterized by behavioral dysfunctions, such as PD [17], HD [18], schizophrenia [19], and cerebral frontal damages [20] it might be extremely valuable to identify the possible contribution of CFT impairments on behavioral regulation and experiential learning, to define possible compensating strategies. To this purpose, a comprehensive acknowledgement of those conditions possibly affected by CFT deficits and the understanding of how these difficulties might be related to the clinical picture is recommended.

As illustrated, psychiatric and neurological conditions have been identified as the best candidates for possible CFT impairments [4, 5]. Systematic reviews and meta-analyses have been provided for depression [8] and PTSD [7]; however, no attempt was made to collect and summarize what we know so far about possible CFT difficulties across several different conditions in adulthood. Indeed, such investigation might provide useful guidance for researchers interested in the topic, to be used as a reference point for future works. For this reason, a scoping review was made to provide a narrative overview about the state of art in the field, identify the methodologies adopted, the variety of psychiatric/neurological conditions already considered, and the relevant issues to be addressed by future investigations.

## Methods

The review protocol was developed according to PRISMA extension for Scoping Reviews (PRISMA–ScR; PRSIMA-ScR Checklist is provided in S1 Table) [21]. The final protocol may be available on request. Original articles published in peer-reviewed journals at any time were included if they were written in English and if they quantitively and/or qualitatively measured CFT in adult participants (i.e., > 18 years old) affected by psychiatric or neurological diseases. Records for which the full text was not available were not included, since the impossibility to get detailed information.

Records were identified by searching the most relevant electronic databases in the field: PsycInfo, PubMed, Scopus, and Web of Science. The last search was run on 23rd January 2020. Specifically, we searched the term "counterfactual thinking" or "reasoning" or "thought" in combination with the following terms: "disease", "illness" "clinical population", "clinical sample", "impairment", "deficit", and "patient". The detailed path used in each database is reported in S1 Appendix.

One reviewer performed the search and made a preliminary eligibility check based on titles and abstracts after removing duplicates. Then, the full text of all the other papers was analyzed in detail, and eligibility was discussed with a second reviewer. If needed, disagreements were resolved by consensus and discussion. Furthermore, the reference lists of the full texts considered were screened to identify additional pertinent articles.

Data were collected by one reviewer according to an extraction sheet that was previously developed with another reviewer. Specifically, the following information was collected for each work: i) characteristics of the sample including disease type and sample size, ii) the tasks administrated and iii) the main results. All data sought were found in the original articles, no additional research was needed.

First, we will provide a brief description of the tasks used and the clinical conditions considered in the reviewed studies that may rapidly inform researchers who want to extend the previous evidence in this field. Then, we will report the main findings grouping the records by the condition considered, that is psychiatric or neurological. Moreover, in each of these two

sections studies will be further grouped according to the specificity of the diseases. More specifically, distinguishing between studies on schizophrenia, depression, post-traumatic stress disorder (PTSD), anxiety and obsessive-compulsive disorders, autism spectrum disorders, and personality traits; concerning the neurological section, involving patients with cerebral lesions, neurological syndromes, or neurodegenerative disorders.

## Results

The selection process is reported in Fig 1. The search performed on PsycInfo, PubMed, Scopus, and Web of Science provided 252 citations. Additionally, 15 studies were detected by manually inspecting the reference lists of the full text papers analyzed. After the removal of

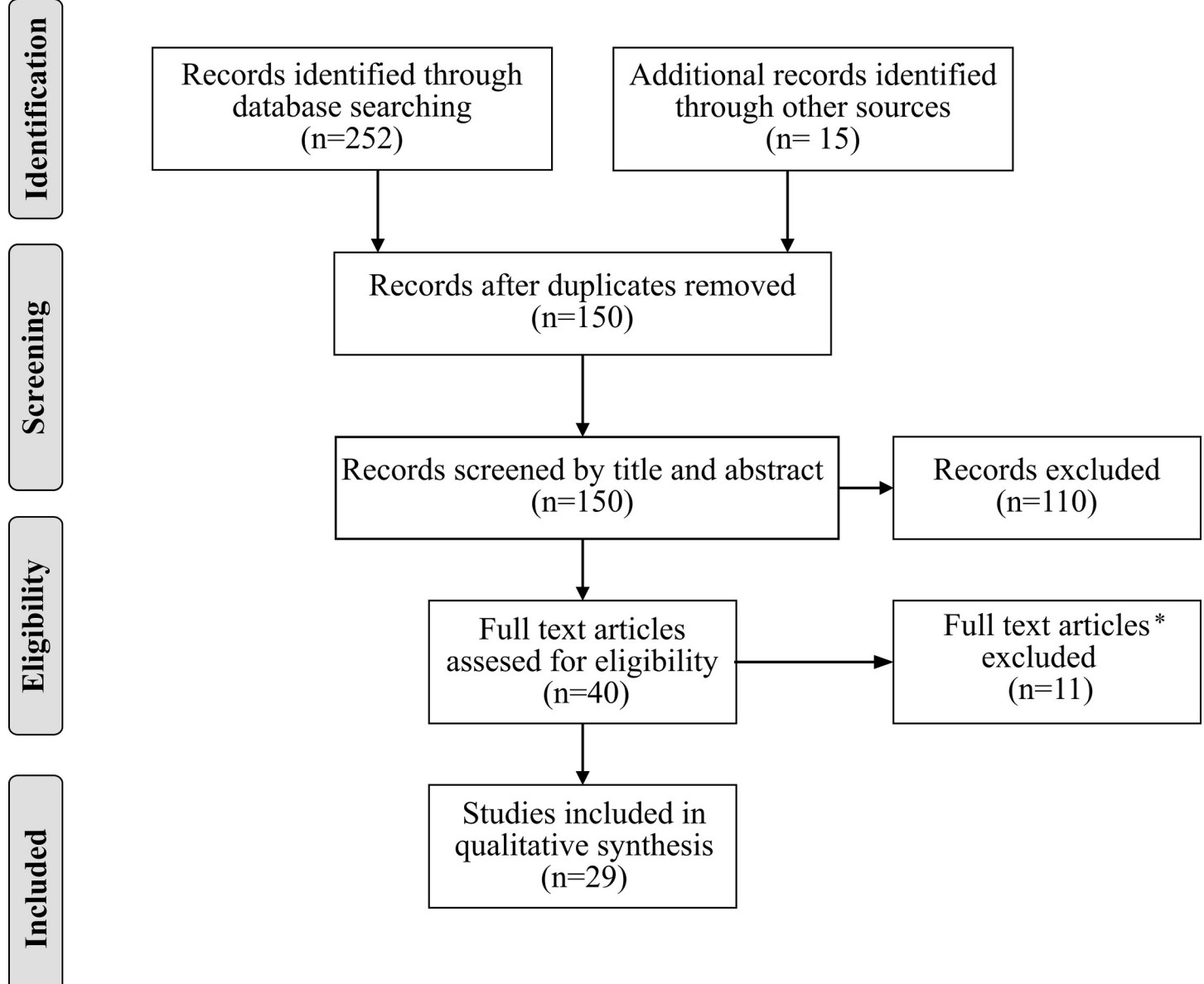

**Fig 1. The flow diagram represents the studies selection process.** * Six studies were excluded since the full text was not available and five were excluded since they did not measure CFT.

duplicates 150 citations remained. All of them were screened for eligibility by title and abstract. A total of 110 records were excluded since eligibility criteria were clearly violated (e.g., studies on children, or different clinical conditions, or not written in English). The full text of 40 works was analyzed in detail. Six studies were excluded since the full text was not available and five were excluded since they did not measure CFT. A total of 29 studies were identified for inclusion in the review. Considering the financial support received, fourteen studies (48%) were founded by public agencies, two (7%) were supported by private foundations, five (17%) received miscellaneous support by both private and public agencies, one (4%) received no support and seven studies (24%) did not specify information about financial support.

The clinical condition considered in each study, sample size, task administered, and main results are reported in Table 1. Twelve studies (41%) probed the **generation of counterfactual thoughts** related to a negative real-life [14–16, 22–27] or a fictional event [9, 23, 25, 28, 29]. Ten studies (34%) administered the **Counterfactual Inference Test** (CIT) [22] to investigate the ability to use CFT to make causal inferences in fictional social scenarios [12, 14–16, 22, 24, 28–31]. Six works (21%) used the **Counterfactual Thinking for Negative Events Scale** [32, 33] or similar questionnaires [34–37] investigating the content (e.g., upward/downward) and the affective aspects of CFT. Four studies (14%) investigated CFT indirectly through a **counterfactual gambling task** [38] measuring the experience of regret in those circumstances in which participants win less (or loss more) than how much they could have won (or lost) making a different gambling choice [11, 12, 31, 39]. Indeed, the experience of regret is a marker of an efficient CFT since it grounds on the ability to figure out how things could have gone differently if a dissimilar choice had been made. Likewise, two studies (7%) [15, 24] investigated the **influence of anticipated counterfactual regret on behavior** [40] asking participants to say how they would have behaved in regret-inducing imaginary circumstances. Individuals who implement an effective CFT are expected to behave anticipating future regret. Finally, two studies (7%) probed the **"causal order effect"** [41], meaning the tendency to choose the first event of a sequence as the most decisive for the final negative outcome [9, 28]. Finally, two studies (7%) investigated the influence of CFT on the automatic activation of behavioral intentions using the **sequential priming paradigm** [10, 30].

Overall, 21 out of 29 studies (72%) focused on psychiatric diseases whereas only eight studies (28%) considered neurological conditions. More specifically, six works focused on **schizophrenia** [9, 10, 12, 22, 28, 30], five probed CFT in the presence of **depressive symptoms** [13, 23, 26, 27, 32] and five investigated the relationship between CFT and **post-traumatic distress** [25, 32–34, 42]. Three studies addressed the relationship between CFT and **anxiety traits** [32, 35–37]. One focused on the **obsessive-compulsive disorder** [11], one on **autism spectrum disorders** [31], and one probed CFT in miscellaneous **psychiatric conditions**, including schizoid and schizotypal traits [32]. Finally, one work probed CFT in participants with different levels of **psychopathy** [43]. As concern neurological diseases, solitary studies focused on **prefrontal cortex** (**PFC**) [16] **orbitofrontal cortex** (**OFC**) [38], and **hippocampal** damages [29], **Parkinson's** [14] and **Huntington's** [15] **diseases**, **relapsing-remitting multiple sclerosis** [39], **Tourette syndrome** [24], and **chronic pain** [44].

## Psychiatric diseases

**Schizophrenia.** The studies reviewed reported a reduced spontaneous generation of counterfactuals thoughts in schizophrenia for both personal [22] and non-personal [9, 28] negative events. On the contrary, composite results have been reported for the ability to use CFT to make inferences, when assessed with the CIT. In two works patients with schizophrenia obtained significantly lower score at the CIT, suggesting a generalized impairment [12, 22].

**Table 1. Characteristics of studies included in the review.**

| | Author | | Clinical Condition | Sample Size | | Tasks | Main Results |
|---|---|---|---|---|---|---|---|
| | | | | Experimental Group | Control Group | | |
| **Neurological diseases** | Andersson and Hovelius, 2006 | | chronic pain | 121 | - | "if only . . ." sentences | generation of counterfactuals with no beneficial effect |
| | Beldarrain et al., 2005 | | PFC damage | 18 | 26 | CFT generation (personal) | fewer spontaneous upward counterfactuals in patients |
| | | | | | | Cued CFT | |
| | | | | | | CIT | no other differences between groups |
| | | | | | | CFT questionnaire | |
| | Camille et al., 2004 | | OFC damage | not specified | not specified | counterfactual gambling task | lower regret sensitivity and lower anticipatory behavior in patients |
| | Mullally et al., 2014 | | hippocampal damages | 6 | 10 | CFT generation (non-personal) | no differences between groups |
| | | | | | | CIT | |
| | McNamara et al., 2003 | | Parkinson's disease | 24 | 15 | CFT generation (personal) | fewer counterfactuals in patients |
| | | | | | | CIT | lower CIT score in patients |
| | Simioni et al., 2012 | | relapsing-remitting multiple sclerosis | 72 | 38 | counterfactual gambling task | lower regret sensitivity and lower anticipatory behavior in patients |
| | Solca et al., 2015 | | Huntington's disease | 24 | 24 | CFT generation (personal) | fewer counterfactuals in patients |
| | | | | | | CIT | |
| | | | | | | IACRB | lower CIT score in patients |
| | | | | | | | comparable IACRB between groups |
| | Zago et al., 2014 | | Tourette's syndrome | 48 | 46 | CFT generation (personal) | no differences between groups |
| | | | | | | CIT | |
| | | | | | | IACRB | |
| **Psychiatric Diseases** | Albacete et al., 2017 | | schizophrenia | 78 | 84 | CFT generation (not personal) | fewer spontaneous counterfactuals in patients |
| | | | | | | CIT | |
| | Barliba and Dafinoiu., 2015 | | psychiatric traits | 300 | - | CFT self-report questionnaire | depression and schizotypal symptoms positively correlate with upward CFT |
| | | | | | | | schizoid traits negatively correlate with upward CFT |
| | | | | | | | PTSD positively correlates with downward CFT |
| | | | | | | | depression negatively correlates with downward CFT |
| | Baskin-Sommers et al., 2016 | | psychopathy traits | 62 | - | counterfactual gambling task | higher psychopathy is not related to a lower regret sensitivity, but participants do not show anticipatory behavior |
| | Blix et al., 2016 | | PSTD | 100 | - | CFT self-report questionnaire | higher frequency of intrusive CFT is related to higher PSTD symptoms |
| | Chase et al., 2010 | | depression | 23 | 23 | counterfactual gambling task | lower regret sensitivity but equal anticipatory behavior in patients |
| | Contreras et al., 2016 | | schizophrenia | 40 | 40 | CFT generation (non-personal) | fewer spontaneous counterfactuals in patients |
| | | | | | | CIT | |
| | Contreras et al., 2017 | | schizophrenia | 37 | 37 | sequential priming paradigm | no differences between groups |
| | Dalgleish, 2004 | | PTSD | 47 | - | CFT generation (personal) | higher frequency of upward and self-referent thoughts, even when trauma survivors perceive no responsibility for the event |
| | | | | 17 | 19 | CFT generation (personal) | |
| | | | | 30 | 30 | CFT generation (non-personal) | no relationship between CFT and post-traumatic distress |
| | | | | - | 43 | CFT generation (non-personal) | |
| | El Leithy et al., 2006 | | PTSD | 46 | - | CFT generation (personal) | CFT frequency positively correlates with PTSD symptoms and negatively correlates with time since the trauma |
| | | | | | | | CFT fluency positively correlates with the generation of behavioral plans |
| | | | | | | | counterproductive thoughts-control strategies are related to a lower decrease of CFT frequency across time |

*(Continued)*

**Table 1.** (Continued)

| Author | Clinical Condition | Sample Size | | Tasks | Main Results |
|---|---|---|---|---|---|
| | | Experimental Group | Control Group | | |
| Gilbar et al., 2010 | PTSD | 96 | 80 | CFT self-report questionnaire | more frequent upward and downward thoughts in patients |
| | | | | | CFT frequency positively correlates with PTSD symptoms and trauma perception |
| | | | | | upward CFT negatively correlates with time since the trauma |
| Gillan et al., 2014 | OCD | 20 | 20 | counterfactual gambling task | increased aspecific emotional arousal and lower anticipatory behavior in patients |
| Hooker et al., 2000 | schizophrenia | 14 | 12 | CFT generation (personal) | fewer counterfactuals in patients |
| | | | | CIT | lower CIT score in patients |
| Larquet et al., 2010 | schizophrenia and OFC damage | 21 schizophrenics | 20 | CIT | lower CIT score in schizophrenic patients |
| | | 10 OFC damaged | | counterfactual gambling task | lower regret sensitivity and lower anticipatory behavior in OBF and schizophrenic patients |
| Markman and Miller, 2006 | depression | 23 mild symptoms | 21 non-depressed | CFT generation (personal) | lower affective function of CFT in the presence of severe depressive symptoms |
| | | 14 severe symptoms | | | higher frequency of controllable thoughts in the presence of mild depressive symptoms, inducing a higher perception of retrospective control |
| Markman and Weary, 1996 | depression | 60 | 61 | | depressive symptoms are related to more controllable counterfactuals, inducing a higher perception of retrospective control |
| Prokopcakova and Ruiselova, 2008 | anxiety traits | 456 | - | CFT self-report questionnaire | higher anxiety traits are related to more frequent CFT |
| Quelhas et al., 2008 | depression | 54 | 70 | CFT generation (non-personal) | depressive symptoms are related to lower CFT and less preparative function |
| | | 23 | 40 | CFT generation (personal) | |
| Roese et al., 2008 | schizophrenia | 15 | 13 | sequential priming paradigm | CFT does not favor related intentions in patients |
| Ruiselová et al., 2007 | anxiety traits | 113 | | CFT self-report questionnaire | higher anxiety traits are related to more frequent CFT |
| Ruiselová et al., 2009 | anxiety traits | 452 | - | CFT self-report questionnaire | higher anxiety traits are related to more frequent CFT |
| Zalla et al., 2014 | autism spectrum disorders | 12 | 12 | counterfactual gambling task | lower regret sensitivity but equal anticipatory behavior in patients |
| | | | | CIT | comparable CIT score |

CFT: counterfactual thinking; CIT: counterfactual inference test; IARCB: influence of anticipated counterfactual regret on behavior test.

On the other hand, two studies showed a preserved overall ability to make inferences, despite a different qualitative pattern of responses [9, 28]. Importantly, this was observed either in patients with active [9] and remissive [28] symptomatology. More specifically, patients perceived temporal "nearly happened" events [9, 28] and unusual antecedents [9] as less powerful triggers of CFT than healthy controls. On the other hand, they considered spatial "nearly happened" events more upsetting [28] and thus more CFT-inducing. However, the latter studies included a significantly larger number of participants, possibly increasing the statistical power and reliability of results. In any case, the experimental evidence suggests that patients with schizophrenia might attribute causality in a different way, with possible substantial consequences on behaviors. However, this atypicality might not extend to all causal attributions. For instance, when patients had to identify among different alternatives the most prominent causal antecedent of an event, they tended to choose the first of the list, showing the typical "causal order effect" [41] reported also in healthy controls [9, 28]. Nevertheless, patients were significantly more frequently unable to perform this task, meaning that some of them could not choose any event at all [9].

Concerning the beneficial effect of CFT in prompting intentions and future behaviors, previous studies using the sequential priming paradigm reported incongruent results. Roese and

colleagues' (2008) observed an impaired ability to create an efficient cognitive link between CFT and behavioral intentions in schizophrenia. Conversely, Contreras and colleagues (2017) reported a comparable capacity of patients with schizophrenia and healthy controls to take advantage of CFT to prompt related behavioral intentions. However, we speculate that the latter results might be more reliable since the authors used an improved version of the paradigm, performed more sophisticated statistical analyses, and included a larger sample of participants.

Nevertheless, the link between CFT and behaviors in schizophrenia might be weakened in more ecological scenarios in which patients have to spontaneously and implicitly translate CFT into efficient behaviors. For instance, in a counterfactual gambling task, participants with schizophrenia with positive symptoms reported lower feelings of regret (denoting an impoverished CFT) and a diminished tendency to modulate their gambling behavior in order to minimize future negative feelings [12]. Furthermore, the effect of an impaired CFT might also be extended to other real-life domains, such as social behaviors. Indeed, Hooker and colleagues (2000) demonstrated that the effect of schizophrenia on social functioning was partially mediated by CFT even when controlling for age and sex. This evidence agrees with the idea that CFT might be a crucial cognitive process supporting an effective psychosocial functioning in the general population [5].

Why should CFT be impaired in schizophrenia? The frontal-executive dysfunction reported in this clinical population [45] might have a crucial role [9, 22]. However, only one study provided solid evidence in this sense [12]. A lack of formal thought and a problematic comparison between hypothetical and actual events might also affect CFT in schizophrenia, as well as an altered affective response (either flat or inappropriate) to negative events [22]. Future studies might attempt to disentangle these issues.

To sum up, the mentioned studies support the hypothesis of impaired conditional reasoning in schizophrenia which might affect the possibility to activate alternative representations of reality in every-day life with significant consequences on patients' behaviors.

**Depression.**   Students with moderate depressive symptomatology (assessed by the Beck Depression Inventory—BDI) generated fewer counterfactuals than non-depressed students when recalling a negative personal event but not a fictional one [23]. Furthermore, depressed students felt less prepared to face a similar (personal) event in the future after the counterfactual generation, suggesting a significantly weaker CFT behavioral regulating function [23]. On the other hand, the effect of CFT on the event-related emotional arousal was comparable in both groups: all participants improved their mood after counterfactual reasoning [23]. Yet, CFT seemed to have a weaker healing effect in individuals with a more severe depressive symptomatology, possibly increasing the perceived distress [27]. To sum up, more depressed students seemed less inclined to reason counterfactually and they may benefit less from CFT than non-depressed students both emotionally and behaviorally.

Nevertheless, CFT might also be beneficial in the presence of depressive symptoms. For example, moderately depressed students tended to produce more counterfactuals that focus on controllable antecedents, improving their feeling of control over the specific event and attenuating their perception of overall control loss [26]. However, this compensatory mechanism might not be sufficiently powerful in the presence of a more severe depressive symptomatology, in fact reducing the feeling of control [27]. Furthermore, other aspects of depression might mediate the tendency to focus on controllable antecedents, influencing the contents of CFT. Specifically, this might be the case of guilty. In other words, depressed individuals would focus on controllable self-related antecedents to make amend for their past behaviors [26].

However, the mentioned works involved undergraduate students with moderate or depressive symptomatology who did not constitute a proper clinical sample. Then, can these results be generalized? Chase and colleagues (2010) assessed the experience of regret in patients

affected by a **major depressive disorder** using a counterfactual gambling task. They reported that patients with depression, especially those with a higher level of self-reported apathy, experienced significantly less regret than healthy controls. Conversely, no differences were found for the experience of joy, disappointment, and relief suggesting that the present findings cannot be explained by a generalized blunting of the emotional arousal. On the contrary, since the feeling of regret is crucially linked to CFT, this result supports the hypothesis of altered counterfactual reasoning in depression, which might be mediated by apathy. Furthermore, severe depression predicts more pessimistic and fewer optimistic counterfactuals [32]. Importantly, the reduced experience of regret and the possible altered counterfactual reasoning in depression might be linked to the OFC dysfunction previously reported [13].

**Post-Traumatic Stress Disorder (PTSD).** A higher frequency of counterfactual thoughts was associated with a more pronounced **post-traumatic distress** among victims of non-sexual assault [42], terror attacks [33, 34], and even in people indirectly exposed to a traumatic event [33]. However, this association decreased since the time of the trauma [34, 42].

Upward counterfactuals seem specifically related to post-traumatic distress [25, 34]. Therefore, in traumatized individuals upward CFT might lose its beneficial preparative function, exacerbating the post-traumatic symptomatology [33]. Indeed, even though the number of upward counterfactuals was found to be positively correlated with the generation of behavioral plans there was no proof that these plans were adaptive [42]. Indeed, a more frequent but dysfunctional CFT could lead to maladaptive behavioral intentions if individuals focus on a narrow range of counterfactuals that does not promote positive alternative behaviors and, on the contrary, maintains the memory of the most fearful aspects of the trauma [42]. This might be especially true in those individuals who adopt dysfunctional strategies to control intrusive thoughts (e.g., "thoughts stopping"), maintaining a high frequency of unpleasant CFT even for a long time after the trauma [42].

Two studies reported a positive association also between PTSD and downward counterfactuals [32, 33] with the optimistic point of view provided by downward thoughts possibly contributing to the establishment of an emotional balance, healing the affective consequences of the traumatic event [32]. Moreover, when a huge number of people are involved in a traumatic event (e.g., bombing attack) the generation of downward counterfactuals might express the tendency to produce counterfactuals that restore normality. Indeed, even though a terror attack cannot be considered a normal event, it usually affects many people. Consequently, a (more) negative outcome could have been plausible.

To summarize, post-traumatic distress seems to be associated with a dysfunctional, unpleasant, and more intense CFT. However, the causal relationship between these two components remains unclear. Furthermore, only two studies compared individuals with a clinical diagnosis of PTSD and non-affected individuals [25, 34], whereas the others [33, 42] did not distinguish between clinically relevant and non-relevant PTSD symptoms, thus limiting the interpretation of the results.

**Autism spectrum disorders.** One study investigated CFT in adults with **Asperger's syndrome** (AS) or **high-functioning autism** (HFA) [31]. No differences were found in the ability to use CFT to make inferences in the CIT. On the contrary, in the counterfactual gambling task, patients with AS/HFA showed reduced susceptibility to regretful scenarios than healthy controls matched for age, sex, education, and IQ. Conversely, no differences were found for the experience of joy, relief, and disappointment. Thus, this result might be explained by a reduced ability to process the self-blame component of regret, rather than reflecting a general impairment of the conscious emotional appraisal. However, AS/HFA individuals showed choice behaviors based on the anticipation of regret. Accordingly, the authors suggested that

the level of psychological arousal associated with regretful events was insufficient to elicit a conscious emotional response, but it was adequate to implicitly orient subsequent behaviors.

**Anxiety traits and Obsessive-Compulsive Disorder (OCD).** Higher **anxiety traits** have been associated with a more frequent CFT in female nurses [35–37]. In turn, CFT was more frequently associated with sadness and was perceived as a bigger obstacle for problem-solving [35]. On the contrary, Barliba and Dafinoiu (2015) did not find any association between clinically relevant anxiety and CFT. Future studies should probe further this issue. On the other hand, in the counterfactual gambling task OCD patients showed a diminished tendency to implement CFT, despite a more intense experience of regret [11]. This result might be explained by a lack of top-down cognitive control (a crucial aspect of impulsivity and compulsivity), despite a proper and even enhance CFT.

**Other personality traits.** Higher **schizoid traits** were associated with a lower frequency of upward counterfactuals, which might reduce the preparative function of CFT [32]. **Schizotypal traits** were positively associated with upward counterfactuals, denoting an increasing frequency of pessimistic thoughts [32]. Unexpectedly, **psychopathy** was not associated with a reduced experience of regret in the counterfactual gambling task. Nevertheless, participants' choice behaviors were not influenced by the anticipation of regretful outcomes [43]. Thus, in psychopathy CFT seems preserved and properly elicits regret, however, this is not translated into functional behaviors aimed to avoid regretful outcomes.

## Neurological diseases

**Cerebral lesions.** Four studies investigated the consequences of cerebral lesions on CFT. Patients with **PFC** damages spontaneously generated fewer upward counterfactuals than healthy controls, suggesting that they might be also less inclined to feel regret [16]. Consequently, they might be less prone to learn from their disadvantageous behaviors. In line with this hypothesis, individuals with **OFC** injuries reported lower emotional arousal than healthy controls in regret-inducing scenarios and they did not modulate their gambling choices in order to minimize the future experience of regret in a counterfactual gambling task [12, 38].

To sum up, fronto-executive cerebral areas seem crucially involved in CFT that, if impaired, can negatively affect individuals' choice behaviors. Furthermore, a lack of spontaneous counterfactual reasoning might also affect the ability to figure out what people think or expect, leading to impaired social interactions [22]. However, the improvement of CFT in these individuals might not be effortless since patients showed a lack of insight into their own ability to think counterfactually [16].

On the other hand, fronto-executive cerebral areas might not be decisive for the use of CFT to make causal inferences [16]. Furthermore, in addition to frontal and prefrontal cortices, it has been proposed that CFT involves also areas related to memory and more specifically the hippocampus [46, 47]. However, Mullally and colleagues (2014) reported that patients affected by retrograde and anterograde amnesia due to **bilateral hippocampal damage** successfully generated counterfactuals about a non-personal event and they properly used CFT to make inferences (CIT). Therefore, and not surprisingly, the hippocampus might be recruited only during episodic real-life CFT [47].

**Neurological and neurodegenerative diseases.** Patients affected by **Huntington's** (HD) [15] and by **Parkinson's** (PD) [14] **diseases** produced fewer spontaneous counterfactuals after recalling a personal negative event than healthy controls. Moreover, they showed a reduced ability to use CFT to make inferences in the CIT. Similarly, CFT seems to be altered in **relapsing-remitting multiple sclerosis** (RRMS) since patients experienced weaker regret than healthy controls in a counterfactual gambling task [39]. In line with these results, RRMS

patients did not modulate their behavior by anticipating possible regretful choices. However, HD patients chose behaviors that minimize the future experience of regret (in the task evaluating the influence of anticipated counterfactual regret on behavior), suggesting a preserved ability to use CFT to positively influence future behaviors. Thus, CFT seems impaired in the neurodegenerative disorders considered, although the effect of this deficit on the behavioral regulation might be differentiated according to the specificity of the disease.

Concerning the etiology of CFT deficits in neurodegenerative diseases, the previous studies point to the specific role of poor executive functioning. This deficit would be likely determined by the progressive deterioration of crucial cerebral areas such as the frontal-subcortical regions in HD and PD [14, 15] and the OFC and the dorso-lateral PFC in RRMS [39]. In line with this hypothesis, an efficient CFT (measured by the Counterfactual Generation Test, the CIT, and a specific task evaluating the influence of anticipated counterfactual regret on behavior) was reported in a sample of patients affected by **Tourette's syndrome** [24] with relatively preserved working memory and executive functioning.

Finally, women affected by **widespread chronic pain** [44] frequently produced counterfactuals involving uncontrollable antecedents that do not provide more efficient alternative behaviors, reducing dramatically the beneficial effect of CFT. However, the absence of a control group and the use of a non-standardized task limit the interpretation of such results.

## Discussion

A scoping review was conducted to provide a narrative overview about previous evidence on CFT in adults with psychiatric or neurological diseases, identifying the methodologies adopted, the variety of psychiatric/neurological conditions already considered, and the relevant issues to be addressed by future investigations. Overall, twenty-nine studies have been reviewed. The generation of counterfactual thoughts related to a negative real-life [14–16, 22–27] or a fictional event [9, 23, 25, 28, 29] and the CIT [12, 14–16, 22, 24, 28–31] were the most popular tasks adopted, followed by the Counterfactual Thinking for Negative Events Scale [32, 33] or similar questionnaires [34–37] and the counterfactual gambling task [11, 12, 31, 39]. Only a minority of studies investigated the influence of anticipated counterfactual regret on behavior [15, 24], the "causal order effect" [9, 28], and the influence of CFT on the automatic activation of behavioral intentions using the sequential priming paradigm [10, 30].

Overall, a wide variety of clinical conditions were investigated; however, most studies focused on **psychiatric diseases** and specifically schizophrenia, depression, and PTSD. Concerning patients with schizophrenia previous findings point to the possible role of reduced CFT [9, 22, 28], low regret sensitivity [12], and qualitatively different causal attributions in favoring suboptimal behaviors [9, 10, 22, 28]. Considering depression, results are a bit more confusing. On one side, it emerged that depressed individuals might be less inclined to reason counterfactually [13, 23] and that they might not take advantage of CFT either to prepare for future similar situations [23] or as an affective regulator [27]. However, other studies reported that depressed individuals generate more pessimistic and fewer optimistic counterfactuals [32] and that they focus significantly more on self-related controllable aspects of events; possibly, empowering their low perception of personal control but, also, incrementing the perceived distress when depression is severe [26]. Thus, these findings only partially agree with the results of Broomhall and colleagues' meta-analysis (2017), showing that a higher predisposition to upward counterfactual thinking and regret "may serve as precursor to depression". Nevertheless, their systematic search enabled to identify more studies than those discussed in the present work, for instance including research in which the evaluation of depressive symptoms was a secondary outcome (e.g., in terminally ill patients or early motherhood). However,

methodological factors (e.g., sample type, instruments used) likely affect the observed relationship between CFT and depression, as they reported. Therefore, the inclusion of additional studies might explain dissimilar conclusions. Furthermore, they considered only upward counterfactuals, whereas no distinction was made in the present work. On the other hand, concerning CFT in PTSD, our report is in line with a recent meta-analysis on the topic [7] suggesting that in PTSD a high frequency of intrusive and dysfunctional counterfactuals about the trauma might exacerbate post-traumatic distress [25, 33, 34, 42]. Thus, therapists should be acknowledged of the possible role of impaired CFT in sustaining post-traumatic symptoms. Finally, a minority of studies assessed CFT in other psychiatric conditions or in the presence of specific personality traits, reporting anomalous CFT generation associated with schizoid and schizotypal [32] and anxiety traits [35–37]; atypical regret sensitivity in AS/HFA [31] and a reduced tendency to modulate one's choice behavior anticipating future regret in psychopathy [43] and OCD [11]. Furthermore, few and isolated studies considered CFT in **neurological conditions**, showing that patients have a reduced tendency to think counterfactually [14–16] and to experience regret for their choices [38, 39] when the diagnosis involves a fronto-executive dysfunction. Nevertheless, the effect of altered CFT on patients' behaviors [15, 24, 38, 39] and the use of CFT to make causal inferences might be differentiated according to the specificity of the disease [14–16, 24, 29].

To sum up, CFT seems to be altered in several psychiatric and neurological diseases, even though with some specificities across the different clinical conditions, and it possibly affects patients' affections and behaviors. Besides, what emerges clearly is that CFT deficits might be specifically associated with a fronto-executive dysfunction, likely related to damages of the PFC and OFC [12, 13, 16, 38, 39], in both psychiatric and neurological conditions. Indeed, the PFC and, more specifically, the OFC are known to be crucially involved in the executive control [48] and they underpin the simulation of counterfactual alternatives in healthy individuals [46, 47]. Additionally, the OFC supports the integration of emotion and cognition in relation to decision making and behavioral planning [49]. Thus, PFC and OFC damages and the presence of fronto-executive dysfunction might be considered crucial risk factors for impaired CFT.

Nevertheless, possible limitations related to our review process might be noted. As mentioned, we did not perform a systematic qualitative and quantitative investigation; thus, possible relevant studies might be omitted, and results have been reported in a narrative fashion. However, a rigorous methodology was followed according to PRISMA guidelines for scoping review; thus, in line with the standards posed by the international scientific community. Also, we included only studies written in English possibly limiting the coverage of our investigation. Nevertheless, English is adopted internationally by the scientific community; thus, we expected more relevant research to be published in this language. Moreover, we focused specifically on psychiatric and neurological diseases in adults, although CFT impairments might be detected also in other diseases, as well as in children. Therefore, future studies might consider reviewing the evidence about different clinical populations, including developmental conditions.

Concluding, the previous research on CFT in the clinical context deserves credit for highlighting the presence of possible CFT impairments in several psychiatric and neurological populations, also drawing quite strong evidence about the neural networks involved. However, studies sometimes lack methodological rigor. For instance, some works did not include a control sample of healthy participants [32, 33, 42–44], did not involve participants with clinically relevant symptomatology [23, 26, 27, 35–37, 43], or did not clearly distinguish between clinically significant and non-significant symptoms [33, 42]. This might be especially risky since counterfactual biases and suboptimal inferential processes have been reported also in the healthy population [50]. Furthermore, four studies included only female [35–37, 44] and one

included only male [43] participants; however, most of the studies controlled for sex and/or other demographic variables such as age and education. Additionally, even though a wide variety of clinical conditions was investigated, most studies focused on psychiatric rather than neurological diseases and specifically, schizophrenia, depression, and PTSD. Therefore, future research might further explore the possible relationship between CFT and the other conditions in which CFT difficulties have been preliminary reported, such as anxiety, OCD, personality traits, autisms spectrum disorders, and psychopathy. Likewise, considering the paucity of evidence about CFT in neurodegenerative disorders, we might recommend further investigations also in this field. Specifically, we might note that given the crucial role of the fronto-executive functioning, the investigation of CFT should be extended to all those conditions characterized by fronto-executive impairments. For instance, Amyotrophic Lateral Sclerosis (ALS) might be of specific interest since 20% to 50% of ALS patients develop Fronto-Temporal Dementia (FTD) [51, 52], including a lack of executive control [53]. On the other hand, since quite a number of studies have been made on CFT in schizophrenia there might be room for a systematic qualitative and/or quantitative analysis (as done for depression and PTSD). Finally, one might speculate that as CFT plays a role in driving everyday behaviors, it might be crucial also when medical decisions are involved. Indeed, neurological and psychiatric patients likely face complex and thorny medical and therapeutic choices, possibly with relevant medical and legal implications. However, the possible effect of impaired CFT on decision making in the clinical context has never been investigated. Future studies might address this issue. Furthermore, future research should focus on the development of interventions aimed at compensating for patients' possible CFT difficulties.

## Supporting information

**S1 Appendix.**
(DOCX)

**S1 Table. Preferred Reporting Items for Systematic reviews and Meta-Analyses extension for Scoping Reviews (PRISMA-ScR) checklist.**
(DOCX)

## Author Contributions

**Conceptualization:** Sofia Tagini, Silvia Torre, Barbara Poletti.

**Data curation:** Sofia Tagini.

**Supervision:** Vincenzo Silani, Gabriella Pravettoni, Barbara Poletti.

**Writing – original draft:** Sofia Tagini.

**Writing – review & editing:** Sofia Tagini, Federica Solca, Silvia Torre, Agostino Brugnera, Andrea Ciammola, Ketti Mazzocco, Roberta Ferrucci, Barbara Poletti.

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
