## [Decision Letter · Decision Letter 0]

4 Dec 2020

PONE-D-20-29746

Facing Clinical Dilemmas: Counterfactual Thinking in Psychiatric and Neurological Diseases.

A Scoping Review.

PLOS ONE

Dear Dr.Tagini,

Thank you for submitting your manuscript to PLOS ONE. After careful consideration, we feel that it has merit but does not fully meet PLOS ONE’s publication criteria as it currently stands. Therefore, we invite you to submit a revised version of the manuscript that addresses the points raised during the review process.

Please submit your revised manuscript by  the deadline. If you will need more time than this to complete your revisions, please reply to this message or contact the journal office at plosone@plos.org. Please include the following items when submitting your revised manuscript:

We look forward to receiving your revised manuscript.

Kind regards,

Vedat Sar, M.D.

Academic Editor

PLOS ONE

Journal Requirements:

2. In your PRISMA flowchart, please include the specific reasons why articles were excluded.

Reviewers' comments:

Reviewer's Responses to Questions

**Comments to the Author**

1. Is the manuscript technically sound, and do the data support the conclusions?

Reviewer #1: Yes

Reviewer #2: Partly

2. Has the statistical analysis been performed appropriately and rigorously? 

Reviewer #1: Yes

Reviewer #2: I Don't Know

3. Have the authors made all data underlying the findings in their manuscript fully available?

Reviewer #1: Yes

Reviewer #2: Yes

4. Is the manuscript presented in an intelligible fashion and written in standard English?

Reviewer #1: Yes

Reviewer #2: Yes

5. Review Comments to the Author

Reviewer #1: This is a qualitative systematic review study focusing on counterfactual thinking abilities among psychiatric and neurological illnesses. Twenty-nine studies have been included. Review methodology is in good order. The review is organized and well-written. Revisiting the findings from a wide range of disorders is appreciated. This work may inspire future studies. I have only minor concerns below

1. The introduction is well-organized, and explanatory about the CFT concept. However, the rationale of investigating CFT across psychiatric / neurological diseases may be clarified. Authors stated that CFT impairments are associated with both groups of diseases, and may lead decision making problems. I think this rationale is somewhat ambiguous, and should be improved.

2. Counterfactual thinking has previously been reviewed and meta-analyzed in several psychiatric disorders such as PTSD, depression. These studies should be cited in the introduction

https://pubmed.ncbi.nlm.nih.gov/28501706/

https://pubmed.ncbi.nlm.nih.gov/32341763/

3. Please avoid using the term “gender” while implying biological sex. Use the term “sex” instead

4. Please revisit your statements about the link between CFT and decision making. In the lack of evidence showing direct association between CFT and decision making, please avoid speculations such as: line 181-183/366-68. I find decision making concept is distracting in the scope of paper. The introduction and discussion should focus on CFT, and the emphasis on the decision making concept should be omitted.

Reviewer #2: This scoping review aimed to summarize the previous evidence about CFT in neurological and psychiatric diseases. The methodology was written appropriately. However, scoping reviews map the nature and state of present literature they also indicate the gaps in the literature in order to create new hypothesis. In terms of these missions of scoping reviews I am not sure that this review fits in this context.

I could not find it sufficient to induce new discussions in the area. Moreover, the logic behind why we need such a review was not clear to me.

6. PLOS authors have the option to publish the peer review history of their article (what does this mean?). If published, this will include your full peer review and any attached files.

Reviewer #1: No

Reviewer #2: No

---

## [Author Response · Author response to Decision Letter 0]

21 Dec 2020

We thank the Reviewers for their suggestions; their requirements were fulfilled, resulting in a significant improvement of the manuscript quality. We hope we have properly met all the Reviewers’ requirements. Details replies are provided below. The track changes mode was adopted in the manuscript.

Reviewer #1: This is a qualitative systematic review study focusing on counterfactual thinking abilities among psychiatric and neurological illnesses. Twenty-nine studies have been included. Review methodology is in good order. The review is organized and well-written. Revisiting the findings from a wide range of disorders is appreciated. This work may inspire future studies. I have only minor concerns below

1. The introduction is well-organized, and explanatory about the CFT concept. However, the rationale of investigating CFT across psychiatric / neurological diseases may be clarified. Authors stated that CFT impairments are associated with both groups of diseases, and may lead decision making problems. I think this rationale is somewhat ambiguous, and should be improved.

Reply: According with the Reviewer’s suggestion the rationale behind our scoping review has been improved. The possible role of CFT difficulties on affective and behavioral regulation has been discussed in further details, illustrating the hypothetical drawbacks of counterfactual hyperactivation/hypoactivation or hypo-deployment. Literature in support of such phenomena in clinical populations has been reported, showing that psychiatric and neurological conditions seem to be the best candidates for CFT difficulties to occur. Accordingly, the possible clinical implications (e.g., on treatment design) of a comprehensive understanding of CFT across several psychiatric/neurological conditions has been discussed. Thus, the need of inclusive scoping review to be used as reference point for future research has been stated. 

Please refer to page 4-6, lines 58-103: “But why do we think counterfactually? The acknowledgement that a better (upward) outcome could have been achieved elicit regret, guilty and self-blame; the greater the salience of the missed achievement, the worse the judgment [2,3]. Nevertheless, we might learn an important lesson for the next time. Indeed, the most prominent function of upward CFT is to provide useful insight about possible alternative ways to achieve missed goals, influencing future behaviors [4,5]; obviously, as long as we recognize that what we did (or did not) was the causal antecedent of our failure. On the contrary, downward counterfactuals denote worse possible scenarios, prompting a sense of relief, thus, having a beneficial affective-regulation purpose [6]. 

Consequently, impaired CFT might have deleterious effects on the ability to modulate both behaviors and affections. Lack of CFT could impact behavioral regulation preventing individuals’ experiential learning and determining underachievement and/or social dysfunction; conversely, excessive CFT may lead to unnecessary worries, anxiety, and dysphoria [4,5]. Thus, CFT difficulties might be somehow related to all those clinical conditions characterized by intense psychological distress, and emotional or behavioral dysfunctions. In support of this hypothesis, meta-analyses showed that there is a significant association between CFT hyperactivation and post traumatic stress disorder (PTSD) [7] and between upwards counterfactuals generation and depression [8]. Furthermore, altered CFT was observed in other psychiatric conditions, such as schizophrenia [9,10] and the obsessive compulsive disorders (OCD) [11]. The causal direction of these associations and the role of possible moderating factors are still under debate [4,7,8]. On the other hand, literature suggests quite strongly that CFT difficulties might be related to fronto-executive vulnerabilities, at least concerning depression and schizophrenia [4,12,13]. Indeed, CFT impairments have been observed also in neurological conditions characterized by a fronto executive dysfunction, such as Parkinson’s Disease (PD) [14], Huntington’s Disease (HD) [15] and frontal lobe damages [16].

 To sum up, CFT difficulties may be detected in a wide range of psychiatric and neurological conditions, with possible negative effects on people’s affective and behavioral regulation. Accordingly, the acknowledgement of the clinical populations possibly involved might be essential for optimal assistance and proper clinical management. For instance, a more refined understanding about the possible role of counterfactuals in maintaining PTSD and depressive symptoms might be crucial for tailoring efficacious interventions [7], which could address self-compassion and aim at minimizing the contemplation of negative outcomes [8]. Furthermore, in those conditions characterized by behavioral dysfunctions, such as PD [17], HD [18], schizophrenia [19], and cerebral frontal damages [20] it might be extremely valuable to identify the possible contribution of CFT impairments on behavioral regulation and experiential learning, in order to define possible compensating strategies. To this purpose, a comprehensive acknowledgement of those conditions possibly affected by CFT deficits and the understanding of how these difficulties might be related to the clinical picture is recommended. 

As illustrated, psychiatric and neurological conditions have been identified as the best candidates for possible CFT impairments [4,5]. Systematic reviews and meta-analyses have been provided for depression [8] and PTSD [7]; however, no attempt was made to collect and summarize what we know so far about possible CFT difficulties across several different conditions in adulthood. Indeed, such investigation might provide useful guidance for researchers interested in the topic, to be used as a reference point for future works. For this reason, a scoping review was made to provide a narrative overview about the state of art in the field, identify the methodologies adopted, the variety of psychiatric/neurological conditions already considered, and the relevant issues to be addressed by future investigations.”

2. Counterfactual thinking has previously been reviewed and meta-analyzed in several psychiatric disorders such as PTSD, depression. These studies should be cited in the introduction

https://pubmed.ncbi.nlm.nih.gov/28501706/

https://pubmed.ncbi.nlm.nih.gov/32341763/

Reply: We thank the Reviewer for having mentioned this; works have been included in the literature background illustrated in the introduction. 

Please refer to page 4, lines: 71-74 “In support of this hypothesis, meta-analyses showed that there is a significant association between CFT hyperactivation and post traumatic stress disorder (PTSD) [7] and between upwards counterfactuals generation and depression [8].”

3. Please avoid using the term “gender” while implying biological sex. Use the term “sex” instead

Reply: All the occurrences of the term “gender” have been replaced with the term “sex”, as suggested. 

4. Please revisit your statements about the link between CFT and decision making. In the lack of evidence showing direct association between CFT and decision making, please avoid speculations such as: line 181-183/366-68. I find decision making concept is distracting in the scope of paper. The introduction and discussion should focus on CFT, and the emphasis on the decision making concept should be omitted.

Reply: According with the Reviewer’s proposal, speculations have been removed and emphasis on decision making was omitted in the introduction and significantly scaled down in the discussion. Indeed, we believed that a brief mention about the possible link between CFT impairments and medical decision capacity might be an inspiring “food of thoughts” for both clinicians and researchers. However, the speculative nature of our proposal was enhanced. 

Please refer to page 4, lines 51-57: “Properness of our past actions is judged according to counterfactual possibilities; that is, what could have been if we had behaved differently. The ability to simulate alternatives to factual events, is called counterfactual thinking (CFT), and it is a signature of humans’ cognition [1]. Conditional propositions are typically used to think counterfactually, often converting unusual behaviors into more “normal” antecedents [1]: “If I had taken the usual way, I would not have missed the flight”. The closer the outcome is to the expected (missed) goal, the higher the probability to elicit CFT [1]” and page 27 lines 469-475: “ Finally, one might speculate that as CFT plays a role in driving everyday behaviors, it might be crucial also when medical decisions are involved. Indeed, neurological and psychiatric patients likely face complex and thorny medical and therapeutic choices, possibly with relevant medical and legal implications. However, the possible effect of impaired CFT on decision making in the clinical context has never been investigated. Future studies might address this issue. Furthermore, future research should focus on the development of interventions aimed at compensating for patients’ possible CFT difficulties.”

Reviewer #2: This scoping review aimed to summarize the previous evidence about CFT in neurological and psychiatric diseases. The methodology was written appropriately. However, scoping reviews map the nature and state of present literature they also indicate the gaps in the literature in order to create new hypothesis. In terms of these missions of scoping reviews I am not sure that this review fits in this context. I could not find it sufficient to induce new discussions in the area. Moreover, the logic behind why we need such a review was not clear to me.

Reply: According to the Reviewer’s remark, gaps in literature and suggestions for possible future research have been both extended and enhanced in the discussion. 

Please refer to page 27, lines 458-475: “Therefore, future research might further explore the possible relationship between CFT and the other conditions in which CFT difficulties have been preliminary reported, such as anxiety, OCD, personality traits, autisms spectrum disorders and psychopathy. Likewise, considering the paucity of evidence about CFT in neurodegenerative disorders, we might recommend further investigations also in this field. Specifically, we might note that given the crucial role of the fronto-executive functioning, the investigation of CFT should be extended to all those conditions characterized by fronto-executive impairments. For instance, Amyotrophic Lateral Sclerosis (ALS) might be of specific interest since 20% to 50% of these patients develop Fronto-Temporal Dementia (FTD) [52,53], including a lack of executive control [54]. On the other hand, speaking about schizophrenia, given that quite a number of studies have been made in the field, there might be room for a systematic qualitative and/or quantitative analysis (as done for depression and PTSD). Finally, one might speculate that as CFT plays a role in driving everyday behaviors, it might be crucial also when medical decisions are involved. Indeed, neurological and psychiatric patients likely face complex and thorny medical and therapeutic choices, possibly with relevant medical and legal implications. However, the possible effect of impaired CFT on decision making in the clinical context has never been investigated. Future studies might address this issue. Furthermore, future research should focus on the development of interventions aimed at compensating for patients’ possible CFT difficulties.”

Furthermore, as required by Reviewer 2 and as also previously illustrated in detail in response to Reviewer 1 (comment #1), the rationale behind our scoping review has now been improved. The possible role of CFT difficulties on affective and behavioral regulation has been discussed, reporting the relevant literature in support of the specific involvement of psychiatric and neurological conditions. Then, the possible clinical implications (e.g., on treatment design) of a comprehensive understanding of CFT across several psychiatric/neurological conditions has been discussed. Thus, the need of inclusive scoping review to be used as reference point for future research has been stated.

Editorial office:

Reply: Style requirements have been checked and necessary changes have been made.

2. In your PRISMA flowchart, please include the specific reasons why articles were excluded.

Reply: Reasons for exclusion have been included in the Figure caption.

Reply: Supporting Information caption and in-text citation have been provided.

---

## [Decision Letter · Decision Letter 1]

18 Jan 2021

Counterfactual thinking in psychiatric and neurological diseases: a scoping review.

PONE-D-20-29746R1

Dear Dr. Tagini,

We’re pleased to inform you that your manuscript has been judged scientifically suitable for publication and will be formally accepted for publication once it meets all outstanding technical requirements.

Kind regards,

Vedat Sar, M.D.

Academic Editor

PLOS ONE

Additional Editor Comments (optional):

Reviewers' comments:

Reviewer's Responses to Questions

**Comments to the Author**

1. If the authors have adequately addressed your comments raised in a previous round of review and you feel that this manuscript is now acceptable for publication, you may indicate that here to bypass the “Comments to the Author” section, enter your conflict of interest statement in the “Confidential to Editor” section, and submit your "Accept" recommendation.

Reviewer #2: All comments have been addressed

2. Is the manuscript technically sound, and do the data support the conclusions?

Reviewer #2: Yes

3. Has the statistical analysis been performed appropriately and rigorously? 

Reviewer #2: N/A

4. Have the authors made all data underlying the findings in their manuscript fully available?

Reviewer #2: Yes

5. Is the manuscript presented in an intelligible fashion and written in standard English?

Reviewer #2: Yes

6. Review Comments to the Author

Reviewer #2: Authors have addressed all the points that reviewers raised.

There are only a few mistyping e.g. line 119.

7. PLOS authors have the option to publish the peer review history of their article (what does this mean?). If published, this will include your full peer review and any attached files.

Reviewer #2: No

---

## [Editor Report · Acceptance letter]

25 Jan 2021

PONE-D-20-29746R1 

Counterfactual thinking in psychiatric and neurological diseases: a scoping review. 

Dear Dr. Tagini:

I'm pleased to inform you that your manuscript has been deemed suitable for publication in PLOS ONE. Congratulations! Your manuscript is now with our production department. 

Kind regards, 

on behalf of

Dr. Vedat Sar 

Academic Editor

PLOS ONE